



# GeoINR 1.0: an implicit neural representation network for three-dimensional geological modelling

Michael Hillier[1], Florian Wellmann[2], Eric de Kemp[1], Ernst Schetselaar[1], Boyan Brodaric[1], Karine Bédard[3]

[1]Geological Survey of Canada, 601 Booth Street, Ottawa, ON K1A 0E8, Canada

[2]Computational Geoscience and Reservoir Engineering (CGRE) RWTH Aachen University, Mathieustr. 30 52074 Aachen, Germany

[3]Geological Survey of Canada, 490 rue de la Couronne, Quebec City, QC G1K 9A9, Canada

*Correspondence to*: Michael Hillier (Michael.Hillier@nrcan-rncan.gc.ca)

**Abstract.** Implicit neural representation (INR) networks are emerging as a powerful framework for learning three-dimensional shape representations of complex objects. These networks can be used effectively to implicitly model three-dimensional geological structures from scattered point data, sampling geological interfaces, units, and orientations of structural features, provided appropriate loss functions associated with data and model constraints are employed during training. The flexibility and scalability of these networks provide a potential framework for integrating new forms of related geological data and knowledge that classical implicit methods cannot easily incorporate. We present a methodology using an efficient INR network architecture, called GeoINR, consisting of multilayer perceptrons (MLP) that advance existing implicit methods for structural geological modelling. The developed methodology expands on the modelling capabilities of existing methods using these networks by: (1) including unconformities into the modelling, (2) introducing constraints on stratigraphic relations as well as global smoothness along with their associated loss functions, and (3) improving training dynamics through the geometrical initialization of learnable network variables. These three enhancements enable the modelling of more complex geology, improved data fitting characteristics, and reduction of modelling artifacts in these settings, as compared to existing INR frameworks for structural geological modelling. A provincial scale case study for the Lower Paleozoic portion of the Western Canadian Sedimentary Basin (WCSB) in Saskatchewan, Canada is presented to demonstrate the modelling capacity of the MLP architecture using the developed methodology. Modelling results illustrate the method's capacity to fit noisy datasets, represent unconformities, and implicitly model large regional scale three-dimensional geological structures.

## 1 Introduction

Understanding the geometry of the subsurface is of critical importance to wide range of applications including earth resource estimation (e.g., mineral, hydrocarbon, geothermal, groundwater), subsurface storage (e.g., carbon sequestration, radioactive waste), urban planning, climate change, and education. Three-dimensional geological modelling provides a means of



representing the geometry of the subsurface based on available geological point-data, typically from boreholes and outcrop observations, sampling geological units, the interfaces between them, and orientations (planar and linear) of various structural features (Wellmann and Caumon, 2018).

The two most common types of three-dimensional geological modelling approaches are differentiated between explicit and implicit surface representations. Explicit approaches (Caumon et al., 2009; Sides, 1997) employ formulations to directly

characterize three-dimensional surface meshes between geological units/faults and rely on either: (1) digitized wireframes interpreted by users possessing geological expertise – guided by primary geological observations - which are converted into Bézier or NURB curves and surfaces (de Kemp and Sprague, 2003; Sprague and de Kemp, 2005). (2) Minimizing the surface roughness on a carefully constructed initial surface mesh using discrete smooth interpolation (Mallet, 1992, 1997) and supplied geological observations. While these approaches can be used to produce excellent structural models - given sufficient

modelling and geological interpretative skill - they can require extensive time to develop and, are difficult to update and reproduce. Implicit approaches, on the other hand, represent geological surfaces as iso-surfaces in a three-dimensional scalar field, which is interpolated using surface interface points, orientations, and potentially off-surface information (Lajaunie et al., 1997, Frank et al., 2007, Hillier et al., 2014). These approaches directly consider stratigraphic continuities and allow for a more flexible updating process, but give rise to new problems, as they can produce geological models with modelling artifacts

in structurally complex settings. For more details on the different geological modelling approaches, we refer to Wellmann and Caumon (2018).

Classical implicit interpolation, that is non-machine learning estimation, has been thoroughly studied and developed over the last two decades with many extensions and enhancements (Boisvert et al., 2009; Calcagno et al., 2008; Caumon et al., 2012; Cowan et al., 2003; de la Varga et al., 2019; Grose et al., 2019; Grose et al., 2021a; Hillier et al., 2014; Irakarama et al.,

2021; Laurent et al., 2016; Renaudeau et al., 2019; Yang et al., 2019). While their extensions and enhancements are remarkable, the underlying mathematical models by which they have been developed are simply not flexible and scalable enough to be able to incorporate the large amounts of available geological data and knowledge. Inequality constraints (Dubrule and Kostov, 1986; Frank et al., 2007; Hillier et al., 2014), for example, useful for incorporating above and below spatial relationships between geological features (e.g., rock units, geological interfaces) have scalability limitations, as the number of constraints

increase, due to computationally expensive convex optimizations required. Furthermore, modelling in structurally complex settings using sparse, heterogeneously distributed, and noisy datasets remains challenging. In these circumstances, produced models can exhibit modelling artifacts (Hillier et al., 2016; von Harten et al., 2021; Pizzella et al., 2022) that are geologically impossible given the known geological history and spatial relationships between geological features. A common strategy to address modelling artifacts is by adding interpretative points for horizons and faults, curves, or localized surface patches to

implicit interpolants resulting in a hybrid implicit-explicit approach. However, this circumvents the entire philosophy of implicit modelling, namely with respect to their reproducibility and fast modelling results. To construct a useful three-



dimensional geological model for downstream applications requires significant time, and in the end are just one possible realization amongst a family of possibilities. Indeed, there is an infinite set of reasonable geological models that fit the data (Jessell et al., 2010), each of which have varying degrees of uncertainty (Lindsay et al., 2012; Wellmann and Regenauer-Lieb, 2012); some models are more probable over others. With the advent of probabilistic approaches (de la Varga and Wellmann, 2016; Grose et al., 2019), these degrees of uncertainty can be somewhat quantified, but fundamentally rely on the space of models that can be produced from the underlying mathematical models, which do not directly incorporate all available geological data and knowledge. Instead, the variables of these models are varied and optimized to maximize likelihood functions that are chosen and designed to integrate other forms of knowledge and data. In some settings, it's possible that these underlying mathematical models are unable to be reparametrized to conform and respect structural styles and complex relationships known to exist in nature. In addition, the frequency of geologically valid models from the ensemble of models generated from probabilistic approaches may be underrepresented in some settings.

Models tend to converge towards sub-surface reality as more geological data and knowledge is incorporated in the modelling process. For complex geological structures, it becomes increasingly difficult – in comparison to simple structures (e.g., layer cake stratigraphy) – to develop accurate representations. For these scenarios, much more geometric and geological feature relationship information is needed to generate realistic models. Due to the inherent flexibility, efficiency, and scalability of deep learning approaches (Emmert-Streib et al., 2020) to incorporate data and knowledge, they have the potential to provide an ideal framework for incorporating new geological data and knowledge constraints into the modelling process, enabling the modelling of complex geological structures and at scales (e.g., high resolution over mine, regional, and national scales) that were previously unfeasible. Beyond being able to expand on the types of geological constraints for structural modelling in deep learning approaches, they also have potential for direct incorporation of relevant interdisciplinary datasets (e.g., fluid flow, mineralization, $CO_2$ storage) where there exist latent relationships to structural features. Collectively, we see potential for these approaches to provide a needed solution for data and knowledge integration within a single end-to-end manner, and thereby overcome the modelling limitations of existing methodologies, and that more accurate representations of three-dimensional geological structures are efficiently produced.

In recent years there has been increasing interest in deep learning approaches for various geoscience applications including seismic data interpretation (Bi et al., 2021; Perol et al., 2018; Ross et al., 2018; Shi et al., 2019; St-Charles et al., 2021; Wang and Chen, 2021; Wang et al., 2022; Wu et al., 2018; Wu et al., 2019), spatial interpolation of geochemical and geotechnical data (Kirkwood et al., 2022; Shi and Wang, 2021), remote-sensing (Ma et al., 2019), and implicit three-dimensional geological modelling (Hillier et al., 2021; Bi et al., 2022). It is also worth noting the machine learning approach that casts implicit modelling as a multi-class classification problem by Gonçalves et al. (2017). While this is not a deep learning-based approach, it supports continuous implicit modelling but not faulting or unconformities. Although deep learning approaches for implicit three-dimensional geological modelling are promising, they are still in their infancy, and much more research and development





is required for them to reach their full potential. While the recently proposed deep learning approach (Bi et al., 2022) can
generate faulted three-dimensional geological models structurally consistent with the data there are some limitations: it cannot
currently model unconformities, there is ambiguity in how to properly annotate or set scalar constraints on horizon data, and
it may suffer from edge effects that can generate spurious discontinuities.

In this paper, we advance a previous deep learning-based approach for three-dimensional implicit geological modelling (Hillier
et al., 2021) using implicit neural representation networks. In recent years, there has been substantial interest and advancements
in using these neural networks on a wide variety of problems including modelling of discrete signals in audio, image, and
video processing, learning complex three-dimensional shapes, and solving boundary value problems (e.g., Poisson, Helmholtz)
(Sitzmann et al., 2020). Moreover, mathematical connections to kernel methods have emerged (Jacot et al., 2020) to establish
a foundation for numerical analysis. In the field of computer graphics, they are being effectively used to represent complex
three-dimensional shapes (Park et al., 2019; Gropp et al., 2020; Atzmon and Lipman, 2020; Davies et al., 2021; Wang et al.,
2021) and reminiscent of surface reconstruction methods using radial basis function interpolation (Carr et al., 2001). For these
applications, a key advantage for deep learning-based approaches is they can be used to learn the latent space of shapes; a
vector space representing all possible geometrical variations of a specific class of objects (e.g., chairs, cats, folded stratigraphy,
etc). In this space, where objects are represented as points within the space, neural networks can learn common properties
between objects of the same class. Furthermore, points representing objects with similar geometries are closer than points
representing objects with dissimilar geometries enabling the interpolation between objects. For structural geological modelling
applications to leverage this concept, realistic three-dimensional geological simulators are required to generate large training
sets from which implicit geological constraints can be sampled in a manner that mirrors the same characteristics as real-world
data (strong heterogeneously distributed, noisy, with conflicting interpretations). It is important to note that there currently
exists a large training set of idealized synthetic three-dimensional models specifically for the purposes of machine learning
training (Jessell et al., 2022). However, research into sampling strategies for real-world data is needed. In this paper, we focus
on advancing implicit neural representation (INR) network approaches for implicit three-dimensional geological modelling to
support more complex geological structures that will also benefit other deep learning approaches that train on large ensembles
of synthetic geological models. Our aim is to demonstrate INR networks can be used efficiently to incorporate a comprehensive
set of inequality constraints on stratigraphic relations (e.g., knowledge constraints) derived from a stratigraphic column,
support modelling of unconformities, improve data fitting characteristics, and reduce modelling artifacts when modelling
complex geological structures with noisy data.

The remainder of this paper is organized as follows. Section 2 describes the proposed methodology using implicit neural
representation networks for modelling complex geological structures containing unconformities. Section 3 presents modelling





results using the proposed methodology. Section 4 discusses modelling characteristics of the approach, and comparisons with
other approaches. The last section, Section 5, conclusions are given.

## 2 Methodology

### 2.1 Definitions and Notations

To better support the geological relations and feature representations mathematically, we have employed specific symbology.
For clarify, definitions and notations used throughout this paper are provided below.


First, the notations for scalar, vector, set/tuples, and matrix quantities are as follows: lowercase, **bold lowercase**,
UPPERCASE, and **BOLD UPPERCASE**, respectively.

Second, there are three types of geological point data considered in this paper that can sample geological interfaces $I_j$ (e.g.,
either stratigraphic horizons or unconformities), geological units $U_j$, and orientation $O$ (e.g., either planar or linear
measurements). For interfaces $I = (I_0, I_1, I_2, \dots)$, subscripts indicate the chronological order the interface was created, with
smaller integers being older interfaces. For geological units $U = (U_A, U_B, U_C, \dots)$, subscripts also indicate the chronological
order of their formation, with the alphabetical order reflecting the sequence of geological units.

Third, point sets in this paper are denoted by $X$. Subscripts on point sets indicate the specific geological feature the point set
is sampling. For example, $X_{I_0}$ is the point set sampling the geological interface $I_0$.

Fourth, for scalar fields, the following notation is used to shorten expressions. Consider a three-dimensional point $\boldsymbol{x}_j$, let
$\varphi_j^i = \varphi^i(\boldsymbol{x}_j)$ denote the scalar field value associated with the $i$-th scalar field $\varphi^i$ at that point. For a set of points $X_Q$ sampling
a specific geological feature $Q$ let $\varphi_Q^i = \varphi^i(X_Q)$ denote the set of scalar values associated with scalar field $\varphi^i$ at the sampled
points. For the mean scalar field value of a set of points $X_Q$ let

$$\bar{\varphi}_Q^i = \frac{1}{|X_Q|} \sum \varphi^i(X_Q) = \frac{1}{|X_Q|} \sum_{\boldsymbol{x}_j \in X_Q}^{|X_Q|} \varphi_j^i \tag{1}$$

where $|X_Q|$ is the number of elements (e.g., points) in the set $X_Q$. Finally, let the gradient of scalar field $\varphi^i$ at point $\boldsymbol{x}_j$ be
denoted by $\nabla \varphi_j^i$.

## 2.2 Problem statement

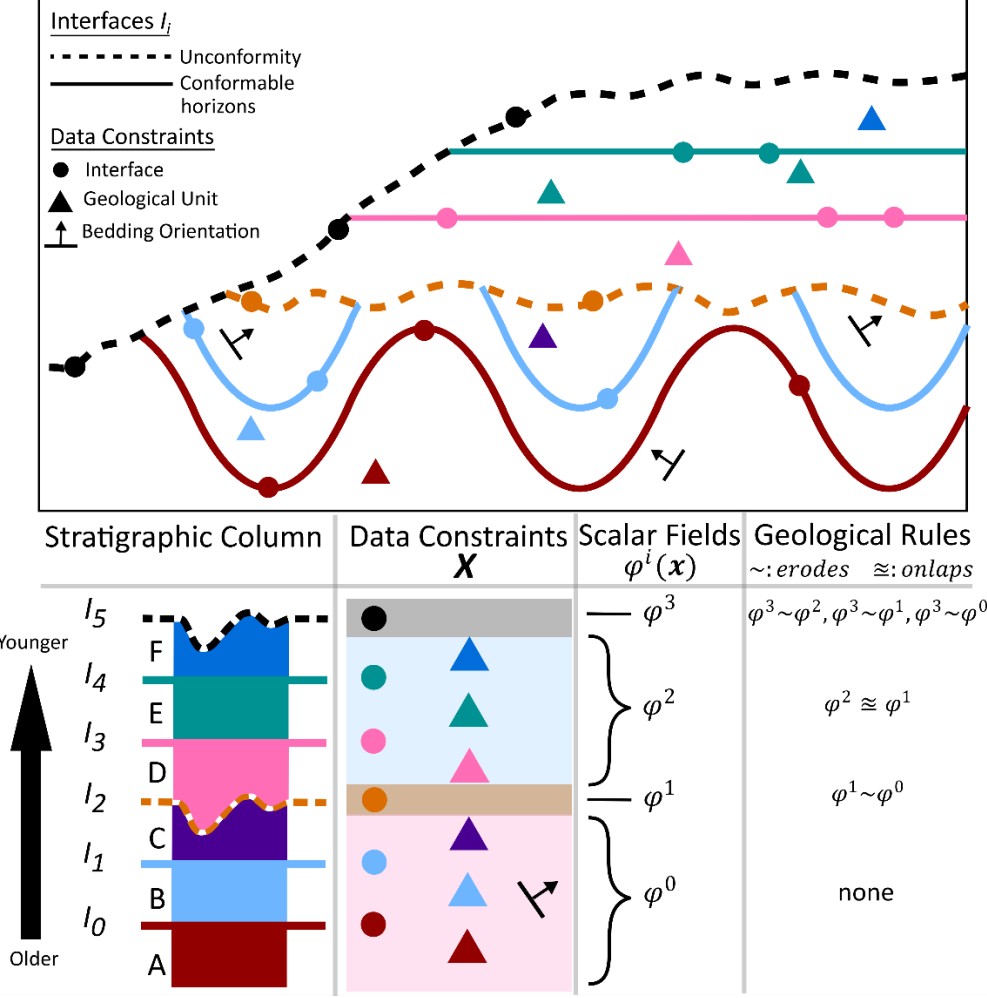

**Figure 1.** Complex geological setting for three-dimensional implicit geological modelling. Inputs for modelling include scattered data constraints, a stratigraphic column, and set of geological rules (erodes ∼, onlaps ≅) for scalar fields $\varphi^i(\boldsymbol{x})$ representing the structures within distinct geological domains.

Our objective is to use MLP neural networks to perform three-dimensional implicit modelling of complex geological settings having both conformable and unconformable structures, given a set of $N$ scattered data points, a stratigraphic column, and set of geological rules as illustrated in Fig. 1. Conformable structures, having undergone the same geological history, exhibit sub-parallel geometries in nearby associated interfaces and strata. In contrast, unconformities are interfaces produced from erosional or halting of sedimentation processes separating strata of different ages - marking a discontinuous transition in the depositional process. Distinct conformable and unconformable structures are modelled separately, each associated with its own implicit scalar field $\varphi^i(\boldsymbol{x})$ and data constraints (Calcagno et al., 2008; de la Varga et al., 2019; Grose et al., 2021b). The



scalar field index $i$ indicates its relative temporal position in the sequence of geological events. Data constraints associated with each scalar field $\varphi^i$ can include points sampling specific sets of geological features such as interfaces $I_k^i$, geological units $U_k^i$, and orientations $O^i$ of interfaces and strata. Subscript $k$ denotes the $k$-th interface or geological unit associated, while the superscript $i$ indicates those geological features are represented by $\varphi^i$. Importantly, the suite of stratigraphic relationships (e.g.,

*above*, *below*, *on*) encapsulated within the stratigraphic column and the geological rules between scalar fields (e.g., erosional, onlap) are incorporated into the modelling process.

Let $\mathcal{M}(X, \varphi, \xi)$ be an implicit model in three-dimensional space where the point set $X = \{\boldsymbol{x}_0, \cdots, \boldsymbol{x}_{N-1}\} \subseteq \mathbb{R}^3$ are the $N$ scattered data points, the tuple $\varphi = (\varphi^0, \cdots, \varphi^{F-1})$ are the $F$ indexed implicit scalar fields, and $\xi$ is a global interpolation constraint (Sect 2.4.5). The global interpolation constraint is to ensure a globally reasonable geological structural model.

Let $\varphi^i\left(X^i, I_k^i, U_k^i, O^i, \mathcal{E}\right)$ be the $i$-th implicit scalar field approximated from the set of points $X^i = \left\{X_{I_k^i}, X_{U_k^i}, X_{O^i}\right\} \subseteq X$ sampling interfaces $I_k^i$, geological units $U_k^i$, and orientations $O^i$ respectively. The scalar field is approximated from the set of





interpolations constraints $\mathcal{E}$ (Sect. 2.4) using these sampled data points. The set of interfaces $I_k^i$ and units $U_k^i$ are arranged in an order of older to younger.

**2.3 Implicit neural representations**

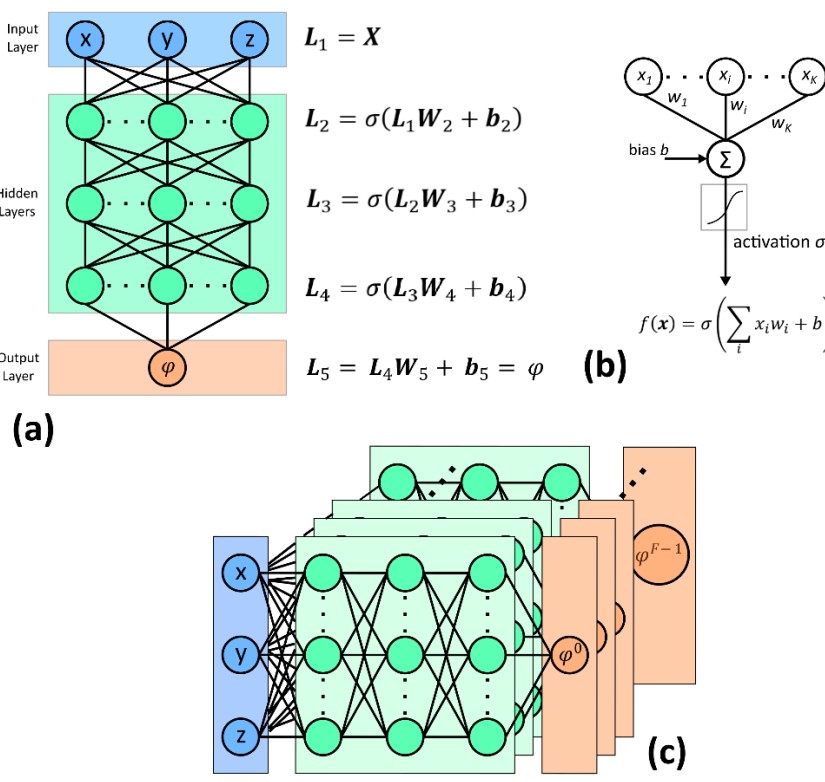

**Figure 2.** Neural network architecture for three-dimensional implicit geological modelling. (**a**) MLP architecture that generates scalar field predictions from spatial coordinates. (**b**) Perceptron neural model and output for a neuron. (**c**) Multiple scalar field predictions for a given point from stacked MLPs in GeoINR network.

Implicit neural representations, also known as coordinate-based representations (Tancik et al., 2020), are neural networks that parameterize implicitly defined functions $\varphi(x)$ where network's inputs $x \in \mathbb{R}^m$ are $m$-th dimensional spatial or spatial-temporal coordinates. These networks typically utilize multilayer perceptrons (MLP), as illustrated in Fig. 2, to learn how to map coordinates into a geometrical representation of shape/structure encoded as an implicit scalar field. Note that other network architectures, such as graph neural networks (Hillier et al., 2021), that learn this mapping are also categorized as INR networks. MLPs are universal approximators capable of approximating any unknown function $f(x)$ provided there are enough hidden neurons (Hornik, 1989). They are composed of three types of layers - input, hidden, and output layers - which transform inputted data into abstract representations and model predictions in the hidden and output layers, respectively. There are three





parameters that define MLP networks: number of hidden layers $N_h$, dimensionality of representations $d_{rep}$, and chosen non-
linear activation function $\sigma$. At every training iteration $t$, errors between the network's outputted scalar fields and interpolation
constraints are measured using developed loss functions presented in the proceeding section. These errors are minimized by
the backpropagation process where the network's variables ($\boldsymbol{W}$'s and $\boldsymbol{b}$'s Fig. 2a) are updated by gradient descent. For complex
geologically settings where there are $F$ distinct conformal and unconformity structures, each associated with a separate implicit
scalar field $\varphi^i$, $F$ MLPs are stacked together resulting in $F$ scalar values being outputted for every point $\boldsymbol{x}$ (Fig. 2c). Following
the training process, multiple scalar fields are combined in a manner respecting the geological rules for erosion and onlapping
of conformal structures onto unconformities (Sect. 2.6)

**2.4 Interpolation constraints and loss functions**

For structural geological modelling, interpolation constraints $\mathcal{E}$ are split into four categories: interface, geological unit,
orientation, and global constraints. For interface and geological unit data, a suite of knowledge constraints on stratigraphic
relations are developed and described in the next section (Sect. 2.4.1). For each constraint type, a corresponding loss function
is developed to accumulate all errors, at every training iteration $t$, measured between the predicted model and set of points for
which a constraint is imposed.

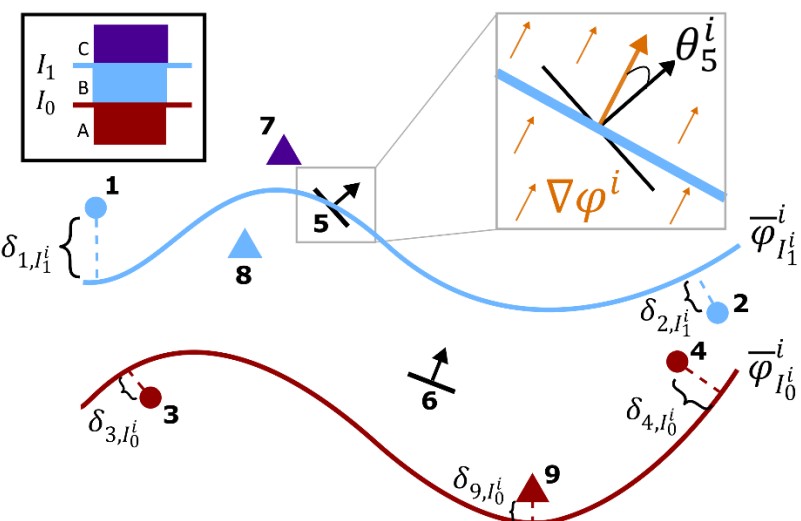


**Figure 3.** Errors associated with interface (circles), orientation (black arrows), and geological unit (triangles) constraints at
training iteration $t$ modelling two conformal interfaces $I_1^i, I_0^i$ and three geological units A, B, C with an implicit scalar field
$\varphi^i$. Approximated signed distances $\delta$ are computed for interface and geological unit data, whereas angular residuals $\theta$ are
computed for orientation data. Insets: (black) stratigraphic column, (gray) angle between scalar gradient (orange) and bedding.



### 2.4.1 Stratigraphic relations and constraints

Stratigraphic relations are defined, in terms of scalar field differences, to encapsulate *above*, *below*, and *on* relationships (e.g., knowledge) between points sampling interfaces and geological units using a given stratigraphic column. From these relations, a suite of constraints for scattered point sets are developed so that the constrained implicit model $\mathcal{M}$ respects the stratigraphic column.

Given a point $\boldsymbol{x}_l \in X_{\Upsilon_j}$ belonging to a point set $X_{\Upsilon_j}$ sampling either a specific interface ($\Upsilon_j = I_j$) or geological unit ($\Upsilon_j = U_j$), a stratigraphic relation is defined as

$$R_{\boldsymbol{x}_l, I_k^i} = \varphi_l^i - \bar{\varphi}_{I_k}^i \tag{2}$$

where $\bar{\varphi}_{I_k}^i$ (Eq. 1) is the iso-value, at training iteration $t$, associated with interface $I_k$ and represented by scalar field $\varphi^i$. The relations indicates whether point $\boldsymbol{x}_l$ is *above*, *below*, or *on* a reference interface $I_k$, modelled with scalar field $\varphi^i$, when the relation value is

$$\begin{aligned} R_{\boldsymbol{x}_l, I_k^i} > 0 \quad &above \\ R_{\boldsymbol{x}_l, I_k^i} < 0 \quad &below, \\ R_{\boldsymbol{x}_l, I_k^i} = 0 \quad &on \end{aligned} \tag{3}$$

respectively. Point sets $X_{\Upsilon_j}$ encoded as stratigraphically above or below an interface $I_k$ are given the following inequality constraints

$$\begin{aligned} R_{\Upsilon_j, I_k^i} = \varphi_{\Upsilon_j}^i - \bar{\varphi}_{I_k}^i > 0 \\ R_{\Upsilon_j, I_k^i} = \varphi_{\Upsilon_j}^i - \bar{\varphi}_{I_k}^i < 0, \end{aligned} \tag{4}$$

respectively. For a point set sampling the reference interface $I_k$ the constraint

$$R_{I_k, I_k^i} = \varphi_{I_k}^i - \bar{\varphi}_{I_k}^i = 0 \tag{5}$$

is used. The complete set of stratigraphic constraints on relations for both interface and geological unit point data illustrated in Fig. 1 are shown in Fig. 4. The set of relations considers interface-interface and unit-interface pairs and are expressed in



matrix form with above relations (yellow) in the upper right and below relations (light purple) in the lower left. For the matrix

of interface-interface relations, *on* relations (green) are along the diagonal. For the *above* relations and associated constraints,

only the ones within distinct geological domains – created by the series of unconformity interfaces – are considered, while the

remaining ones (red) are discarded. These are discarded because points sampling younger geological features can be measured

as being below older modelled interfaces from other geological domains using their associated scalar fields and corresponding

iso-value, depending on their geometries. For example, consider the unconformity interface $I_2$ (Fig. 4), it erodes portions of $I_1$

and therefore the presence of the unconformity can be measured below $I_1$ using $\varphi^0$ (e.g., the scalar field that models $I_1$) in

those portions. This characteristic doesn't apply to any below relations and constraints, since points sampling an interface or

unit must always be below all younger interfaces.

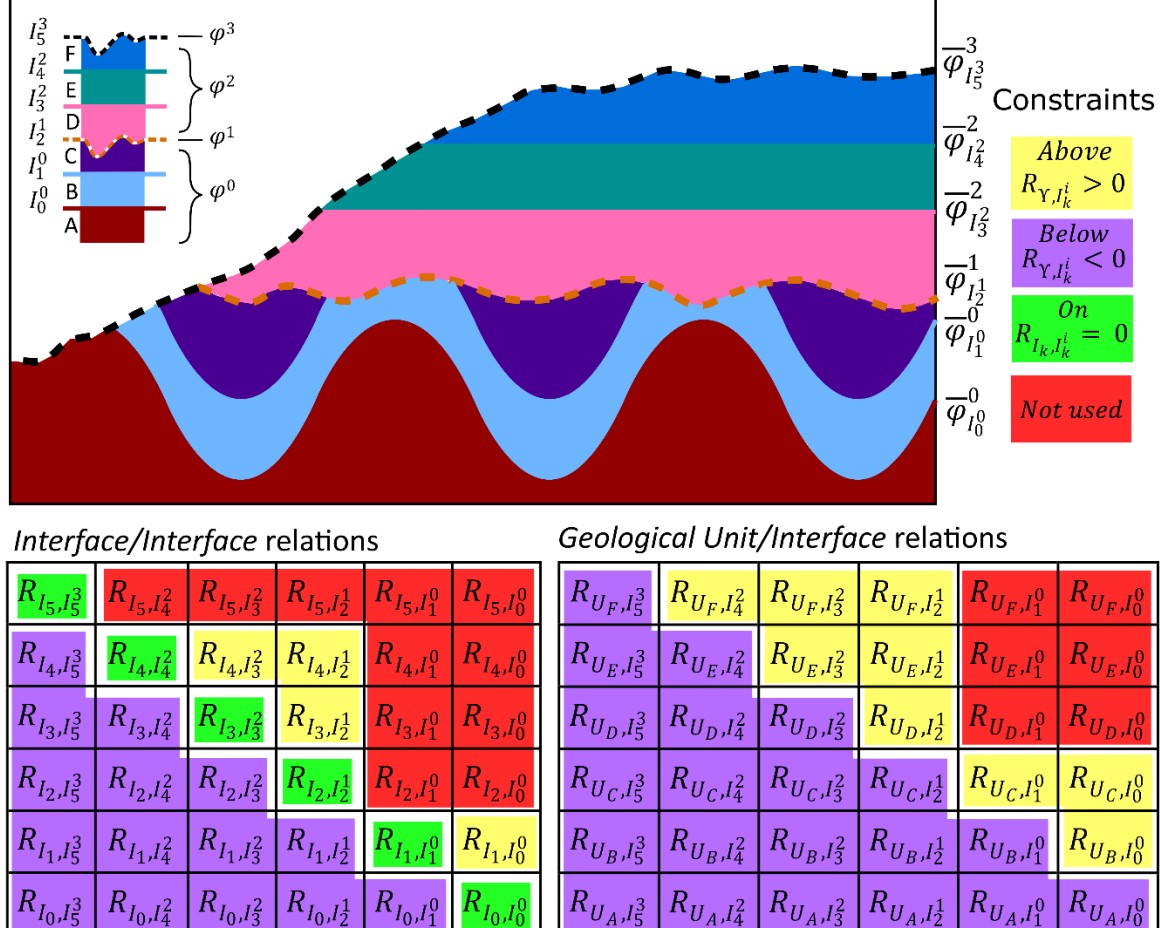

**Figure 4.** Stratigraphic relations between specific interface-interface and geological unit-interface pairs and associated

constraints. Constraints are colored according to their *above* (yellow), *below* (light purple), or *on* (green) spatial relation. For





above relations (upper right matrix block), only the constraints on relations within distinct geological domains are considered

while the remaining constraints are not used (red).

To measure errors at some training iteration $t$ between the implicit model $\mathcal{M}$ at sampled interfaces and geological unit points

$x_l$ and their associated constraints, an approximate signed distance $\delta_{x_l, I_k^i}$ (Caumon, 2010; Taubin, 1994) (Fig. 3) from a

reference interface $I_k$ modelled by $\varphi^i$

$$\delta_{x_l, I_k^i} = \frac{\varphi_l^i - \bar{\varphi}_{I_k}^i}{\|\nabla \varphi_l^i\|} \tag{6}$$

is used. The magnitude of the scalar gradient $\|\nabla \varphi_l^i\|$ in the denominator is an important term to account for changes in unit

thickness between interfaces represented in the scalar field. Smaller magnitudes correspond to thickening of units, while larger

ones are indicative of unit thinning. Consequently, the approximate signed distances are a much more accurate measure of

how far above or below a point is above some reference interface than the scalar differences themselves. This is because scalar

values for various geological features are not meaningful in real-world distances and are not normalized between features.

The three loss functions for the *above*, *below*, and *on* stratigraphic constraints integrating all errors from sets of point sets

$X_\Upsilon$ are given by

$$\mathcal{L}_\Upsilon^{Above} = \sum_{\Upsilon_j \in \Upsilon}^{|\Upsilon|} \frac{1}{|X_{\Upsilon_j}|} \sum_{x_l \in X_{\Upsilon_j}}^{|X_{\Upsilon_j}|} \sum_{I_k^i \in B_{\Upsilon_j}}^{|B_{\Upsilon_j}|} \hat{\delta}_{x_l, I_k^i}, \quad \hat{\delta}_{x_l, I_k^i} = \begin{cases} \dfrac{|\varphi_l^i - \bar{\varphi}_{I_k}^i|}{\|\nabla \varphi_l^i\|} & \varphi_l^i - \bar{\varphi}_{I_k}^i < 0 \\ 0 & \varphi_l^i - \bar{\varphi}_{I_k}^i \geq 0 \end{cases} \tag{7}$$

$$\mathcal{L}_\Upsilon^{Below} = \sum_{\Upsilon_j \in \Upsilon}^{|\Upsilon|} \frac{1}{|X_{\Upsilon_j}|} \sum_{x_l \in X_{\Upsilon_j}}^{|X_{\Upsilon_j}|} \sum_{I_k^i \in A_{\Upsilon_j}}^{|A_{\Upsilon_j}|} \check{\delta}_{x_l, I_k^i}, \quad \check{\delta}_{x_l, I_k^i} = \begin{cases} \dfrac{\varphi_l^i - \bar{\varphi}_{I_k}^i}{\|\nabla \varphi_l^i\|} & \varphi_l^i - \bar{\varphi}_{I_k}^i > 0 \\ 0 & \varphi_l^i - \bar{\varphi}_{I_k}^i \leq 0 \end{cases} \tag{8}$$

$$\mathcal{L}_I^{On} = \sum_k^{|I|} \frac{1}{|X_{I_k}|} \sum_{x_l \in X_{I_k}}^{|X_{I_k}|} |\delta_{x_l, I_k^i}| \tag{9}$$

respectively. Note that $A_{\Upsilon_j}$ and $B_{\Upsilon_j}$ are a set of interfaces $I_k^i$ that are above or below, respectively, a specific geological feature

$\Upsilon_j$ (either an interface or unit). For example, consider the loss function for the below constraints (Eq. 8) associated with the

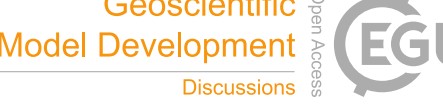

geological unit $U_D$ from Fig. 4. In this case, $Y_j = U_D$ and $A_{Y_j} = A_{U_D} = \{I_5^3, I_4^2, I_3^2\}$ are the set of interfaces above that geological unit. The below constraints for this geological unit require that the points within the set $X_{U_D}$ must be below the interfaces above $A_{U_D}$. If points $x_l \in X_{U_D}$ are above, or $\varphi_l^i - \bar{\varphi}_{I_k^i}^i > 0$, those points will have non-zero errors, otherwise the error will be zero (e.g., respect constraint).

Loss functions associated with above or below stratigraphic constraints are effective in constraining resultant implicit models to respect the sequence provided by a given stratigraphic column. Not only do these loss functions ensure modelled interfaces and strata respect the stratigraphic sequence for each scalar field $\varphi^i$ but also importantly, that they respect the presence of sampled interfaces and strata associated with other scalar fields. To clearly illustrate the latter, consider Fig. 5 where two unconformities are modelled separately with two scalar fields. Without these constraints, unconformities are modelled

independently, a portion of the older unconformity is eroded incorrectly despite the presence of a valid unconformity observation point (e.g., point 1 Fig.5a). With these constraints, all scalar fields are coupled so that the entire geological sequence of all sample interfaces and strata are honored/considered. This resolves an issue in other implicit approaches (Calcagno et al., 2008, de la Varga et al., 2019, Grose et al., 2021b) that treat each scalar field independently. And finally, these constraints help impose the correct scalar field polarity – e.g., the alignment of the gradient of the scalar field $\nabla\varphi$ with

younging direction (direction of younger stratigraphy) - even in circumstances where there are no bedding observations available. Having the correct scalar field polarity is critical in assigning geological domains so that multiple scalar fields can be combined into a resultant scalar field respecting geological rules (erosional and onlap), as well as assigning geological units to modelled volumes (Sect 2.6).

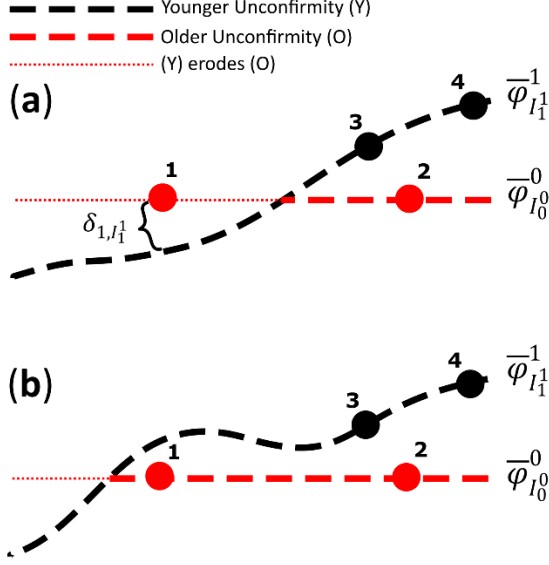






**Figure 5.** The effect of above and below stratigraphic constraints in coupling two scalar fields $\varphi^0$ and $\varphi^1$ modelling two unconformities. (**a**) Without using the constraints and (**b**) with using the constraints.

**2.4.2 Interface constraints**

For interface data, there are four interpolation constraints. Firstly, the variance of all scalar field values $\varphi^i_{I^i_k}$ on a sampled interface $I^i_k$ are roughly zero

$$Var\left(\varphi^i_{I^i_k}\right) = 0. \tag{10}$$

This iso-value constraint ensures that the scalar field at the sampled locations for $k$-th interface $X_{I_k}$ are the same and has the following associated loss function

$$\mathcal{L}^{var}_I = \sum_{I_k \in I}^{|I|} Var\left(\varphi^i_{I^i_k}\right). \tag{11}$$

The other three constraints utilize the stratigraphic relations to enforce the above, below, and on constraints. Combined, the resulting loss function for interface data is

$$\mathcal{L}_I = \mathcal{L}^{var}_I + \mathcal{L}^{On}_I + \mathcal{L}^{Above}_I + \mathcal{L}^{Below}_I. \tag{12}$$

The first two loss functions both constrain the implicit model to respect the locations of sampled interfaces, while the last two ensure that the sequence of sampled interfaces respects the given stratigraphic column.

**2.4.3 Geological unit constraints**

To constrain the implicit model with geological unit data $U$, the above and below stratigraphic constraints are applied to this dataset. Consequently, the resulting loss function for geological unit data is

$$\mathcal{L}_U = \mathcal{L}^{Above}_U + \mathcal{L}^{Below}_U. \tag{13}$$





### 2.4.4 Orientation constraints

For an orientation data point $\boldsymbol{x}_j \in X_{O^i}$ associated with a scalar field $\varphi^i$, an angular constraint $\theta_j^C$ characterizes the angle between the orientation vector $\boldsymbol{v}_j$ and the scalar gradient $\nabla\varphi_j^i$ at $\boldsymbol{x}_j$. For normal data (e.g., bedding orientation with younging direction), the interpolation constraint is

$$\theta^C = 0°, \tag{14}$$

while for tangent data (e.g., lineations, fold axis) it is

$$\theta^C = 90°. \tag{15}$$

The loss function associated with orientation data $\theta^i$ measures angular errors (Fig. 3) between the given angular constraints $\theta_j^C$ and angles $\theta_j^i$ computed from the implicit model $\mathcal{M}$ at some training iteration, and is given by

$$\mathcal{L}_O = \sum_{i=1}^{F} \frac{1}{|O^i|} \sum_{j \in O^i}^{|O^i|} \left| cos\theta_j^C - cos\theta_j^i \right| \tag{16}$$

where $cos\theta_j^i$ is computed from

$$cos\theta_j^i = \frac{\boldsymbol{v}_j \cdot \nabla\varphi_j^i}{\|\boldsymbol{v}_j\|\|\nabla\varphi_j^i\|}. \tag{17}$$

### 2.4.5 Global constraint

It is well established, as previously mentioned in the introduction, that a disadvantage of implicit approaches for structural geological modelling is that they can produce modelling artifacts, commonly referred to as 'bubbly' artifacts, yielding geologically unreasonable models (de Kemp et al., 2017) particularly in complex structural settings. One way to address this problem is to impose a global constraint over the modelling domain using energy minimization principles. Here we use the Eikonal constraint (Gropp et al., 2020), a unit-norm constraint, for this purpose

$$\left\|\nabla\varphi^i(\boldsymbol{x})\right\| = 1 \tag{18}$$





which has the following associated loss function for the implicit model

$$\mathcal{L}_\xi = \sum_i^F \frac{1}{|\Omega_s|} \sum_{x_j \in \Omega_s}^{|\Omega_s|} \left( \left| \left\| \nabla \varphi_j^i \right\| - 1 \right| \right) \tag{19}$$

where $\Omega_s$ are a set of points sampling the modelling domain $\Omega$. Due to the efficiency and computational scalability of MLP neural networks, sufficiently sampling the domain, even densely, is feasible. The effect of the global constraint on the scalar field is that it promotes sub-parallel geometries in nearby strata throughout a modelling domain. The effect is illustrated in the case study (Sect. 3).

**2.4.6 Resultant loss function**

The resultant loss function, or total loss function $\mathcal{L}$, for all geological constraints is simply the sum of the individual loss functions and is given by

$$\mathcal{L} = \mathcal{L}_I + \mathcal{L}_U + \mathcal{L}_O + \lambda \mathcal{L}_\xi \tag{20}$$

where the loss function $\mathcal{L}_\xi$ for the global constraint is weighted by the lambda term, $\lambda > 0$. The larger its value the more scalar fields are smoothed. This function represents the loss landscape (Li et al., 2018), or objective function, given a set of constraints and in which the learning algorithm attempts to find its minimum. The location in which the loss landscape is a minimum corresponds to the set of neural networks variables that yield minimal error between the network's predictions and geological constraints.

**2.5 Training**

An important training aspect to our proposed implicit neural representation networks is a geometrical initialization of network variables. The variables are initialized such that the resulting outputted scalar field represents a shape with reasonable starting geometry for a specific geological application, which will be evolved through training by fitting given data constraints. Using standard variable random initialization schemes (Glorot and Bengio, 2010; He et al., 2015), resulting output scalar fields can

be far from an optimal starting point for training, especially as the network's complexity increases (Fig. 6a). Consequently, if the training algorithm is rerun many times using the same conditions (Fig. 6b), resulting structural models can exhibit large variance in modelled structures. To solve these issues, training starts with a geometrically reasonable scalar field by geometrical initialization of network variables. For stratigraphic modelling, network variables are initialized to produce a

planar geometry (Fig. 6d), whereas for intrusive-like modelling, they are initialized to produce a spherical geometry (Fig. 6c) (Atzmon and Lipman, 2020).

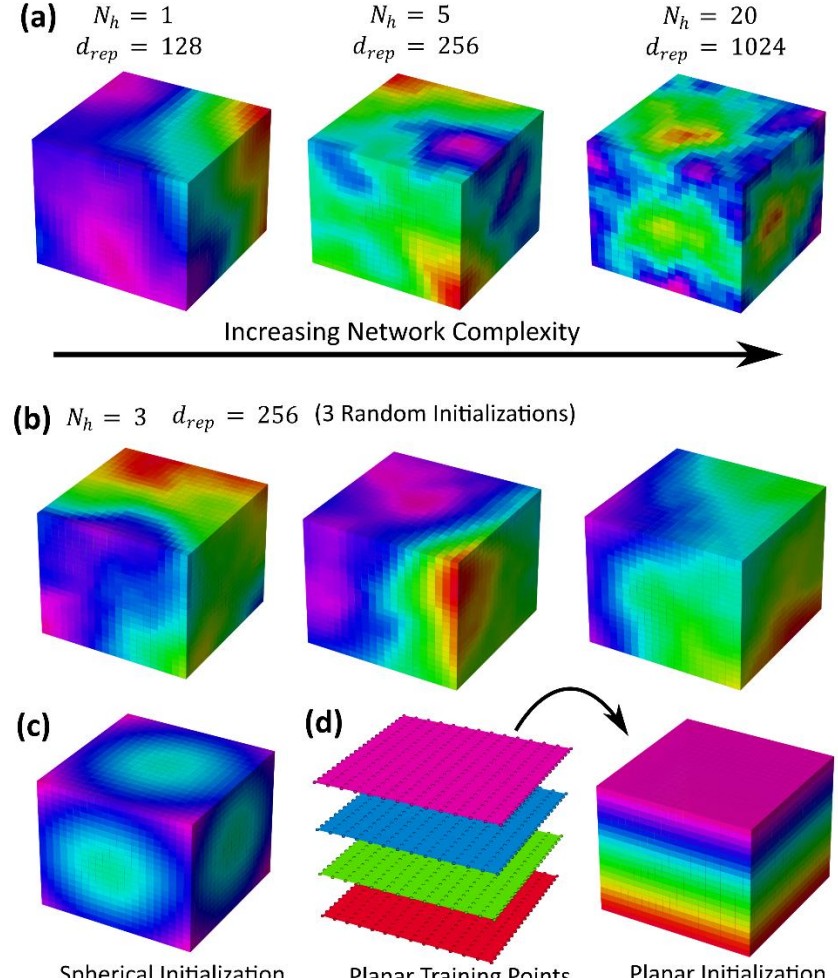

**Figure 6.** Scalar fields generated from initialization of network variables. (**a**) Effect of increasing network complexity on generated scalar field by increasing number of hidden layers $N_h$ and dimension of hidden representations $d_{rep}$. (**b**) Scalar fields generated from three random initializations of network variables. (**c**) Spherical initialization. (**d**) Planar initialization

using pre-trained network applied to points sampling layer-cake volume.

To initialize our networks to produce a scalar field with a planar geometry we first pretrain a MLP network for 1000 epochs with the same parameterization ($N_h$, $d_{rep}$, $\sigma$) on a synthetic dataset densely sampling four layer-cake interfaces (Fig. 6d). The pre-trained network's parameters are saved and loaded into each of the $F$ stacked MLP networks which are updated by training

on an unseen stratigraphic dataset. This can be viewed as transfer learning (Zhuang et al., 2021); applying what is learned for one problem onto a similar problem. An added benefit to using pretrained networks is reproducibility in modelling results





since the network is initialized with the same parameters. Furthermore, the number of training epochs to converge (e.g., minimal losses) is reduced.

Another training aspect utilized in the proposed methodology is applying learning rate schedules in the Adam optimizer
(Loshchilov and Hutter, 2017). Learning rate schedules adjust the learning rate during training by decreasing the rate according to a prescribed schedule. While the Adam optimizer does adapt the initialized learning rate on a per-parameter biases, there is a benefit to decreasing the adaptable learning rate with increasing training epochs. Empirically, we have found that applying either step decay or cosine annealing learning rate schedules yields much lower losses and consequently better data fitting characteristics.

**2.6 Geological domains and combining scalar field series**

After implicit scalar functions are constrained by training, gridded points sampling a geological volume are inputted to the trained MLP network to generate $F$ separate continuous scalar fields, each representing a distinct geological feature, throughout the volume of interest (Fig. 7a). Since unconformity interfaces can erode (e.g., cut) older geological features, there are regions of space where those features are no longer present. To cut portions of a geological feature removed by an
unconformity, its associated continuous scalar field is cut by the modelled unconformity interface. As a result, geological features are partitioned into geological domains (Fig. 7b) where those features are present, and their associated scalar fields and geological units are defined (Fig. 7c). Since a point in the modelled volume can only be associated with a single scalar field, the scalar field series and their associated geological units is combined such that only the domain in which each scalar field and set of units is defined is merged into a resultant scalar field and geological unit model (Fig. 7d).




**Figure 7.** Constructing geological domains and combining scalar fields. (**a**) (top) Scalar field series and modelled interfaces with their regions (bottom) defined from associated inequalities. Scalar fields associated with unconformities have Boolean





masks $M^i$ defining above and below regions. (**b**) Geological domains constructed from Boolean masks. (**c**) Scalar fields and geological units assigned to geological domains. (**d**) Combined scalar fields and geological units.


Geological domains are spatial partitions created by boundaries defining discontinuous features (e.g., unconformities and faults) within which continuous geological features (e.g., conformable stratigraphy) exist. In this paper, only unconformity boundaries are used to create geological domains, although the same idea can be used for faulting. See discussion (Sect. 4) for future work with incorporation of faulting. To construct geological domains, first mask arrays defining above and below (Fig.

7b) regions for each unconformity interface within the modelling volume are computed using the associated scalar field and inequalities. As mentioned previously, the notion of above/below a reference interface defined by an iso-value, also known as polarity, is provided by the scalar gradient (Fig. 7a) that points in the direction of younger stratigraphy (e.g., Younging direction). For example, volumes above and below an interface $I_5$ modelled with $\varphi^3$ are defined, respectively, by

$$
\begin{aligned}
\varphi^3 &\geq \overline{\varphi}_{I_5^3}^3 \\
\varphi^3 &< \overline{\varphi}_{I_5^3}^3
\end{aligned}
\tag{21}
$$


where $\overline{\varphi}_{I_5^3}^3$ is the iso-value associated with $I_5$. The mask array $M^i$ associated with the unconformity is set to *True* wherever above the interface, and *False* below it. Secondly, from the geological rules associated with the unconformity scalar fields, an appropriate set of Boolean logic is applied to the mask array(s) to define the geological domain. For example, consider domain $D^1$ in Fig. 7b, it is defined wherever it is below $I_5$ (where $M^3$ is false; $!M^3$) and above $I_2$ (where $M^1$ is true; $M^1$). Domains for

older geological features have a bigger set of Boolean logic applied to mask arrays since there are more younger unconformities that can erode those volumes as compared to younger geological features. With the geological domains defined, scalar fields and geological units are both assigned to their associated geological domains thereby combining all the scalar fields and geological units into a resultant three-dimensional geological volumetric model.

### 2.7 Iso-surface Extraction

Iso-surface extraction methods can be applied to specific regions of implicit scalar field volumes provided appropriate Boolean masks are given and can be useful for obtaining the geological horizons of conformal domains. However, obtaining unconformity interfaces using this approach will lead the production of anti-aliasing artifacts in triangulated surfaces. To resolve this issue, we develop an algorithm (see Algorithm 1 in Appendix A) using the open-source library PyVista (Sullivan and Kaszynski, 2019) to generate all iso-surfaces that can be cut by unconformities. The algorithm first extracts the set of

continuous iso-surfaces for each of the $F$ scalar fields computed within a gridded volume. Next, the set of iso-surfaces are iterated on, going from oldest to youngest and processed. For a given surface being processed, that surface is progressively cut by younger unconformities above it in the stratigraphic column, again going from older to younger.



## 3. Case Study

Modelling results produced by the proposed methodology for a real-world case study of a sedimentary basin are presented
here to demonstrate proof of concept. The dataset used for this purpose is a compilation of formation tops and unconformity
picks, extracted from March and Love (2014), from the Lower Paleozoic portion of the Western Canadian Sedimentary Basin
(WCSB) in Saskatchewan, Canada. The interface constraint data consisted of 4708 tops and unconformity picks sampling 4
unconformities and 3 conformable horizons. The depths of the picks were interpreted from geophysical well logs and correlated
to core samples when available. Due to the interpretative nature of the constraint data – an attribute for all real-world geological
datasets - the data can be characterized as noisy as their exact positions are uncertain (Fig. 8a (right)). Moreover, this presents
an opportunity to test whether the proposed methodology can be useful for generating three-dimensional geological models
from regional compilation datasets commonly found in Geological Survey Organizations (GSO). In addition to interface
constraint data, augmented data consisting of intraformational units were generated by sampling along well intervals to
demonstrate their modelling impact as compared to using only interface data. This augmented data is not required to produce
a geologically representative model (Fig. 8b) but serves to demonstrate that the methodology successfully handles this type of
data. Intraformational units are sampled along a well interval between two interfaces only if the interfaces are stratigraphically
adjacent. Interpreted depths of successive formation tops (e.g., interfaces) along the well path may not always be
stratigraphically adjacent, either because the top could not be identified or that a portion of stratigraphy was eroded. In these
cases, intraformational units are not sampled along a well interval (Fig. 9). For the case study, sequenced well intervals were
sampled at every 20 meters (vertical resolution of our voxel grid; 5 km was the horizontal resolution) and generated 11270
sampled intraformational units.



**Figure 8.** Modelling results for the Lower Paleozoic portion of the WCSB in Saskatchewan generated using the proposed

GeoINR methodology. Data and modelled results use 100x vertical exaggeration to visualize provincial scale model. (**a**)

Model's geographical coverage, stratigraphic column, formation tops and sampled intraformational data constraints (smaller





spheres). (**b**) Modelled horizons, resultant scalar field and formation units. (**c**) Section view highlighting data fitting characteristics and the effect of removing intraformational units from computation. (**d**) Side view highlighting geometry of unconformities in modelled interfaces and associated resultant scalar field. (**e**) Effect of using global smoothness constraint on
a scalar field.

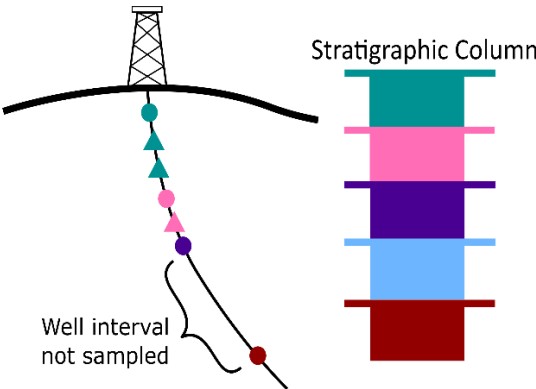

**Figure 9.** Sampling intraformational units (triangles) along well path.


    Organizing the geological point dataset first requires all necessary knowledge to be extracted from the stratigraphic column. Stratigraphic knowledge including the geological rule of the interface (e.g., erosional, or onlap (conformable)) and the set of interfaces above and below each interface and geological unit are tabulated (Table 1 and 2). Corresponding tables describe the set of stratigraphic relations (Fig. 4), and which scalar field series is associated to a particular interface or unit. This information
is used for implementation purposes so that associated loss functions can be computed measuring errors between the stratigraphic constraints and the current version of the model at some training iteration $t$.

    As with any machine learning algorithm, neural network inputs require normalization for the network to learn useful latent representations and yield accurate predictions. Inputs for implicit neural representation networks, which are spatial coordinates
in this case, are normalized to some range for each coordinate dimension. For the proposed network architecture, we normalize each coordinate dimension to range from [-1, 1]. It is worth noting that if the coordinate ranges $(\Delta x, \Delta y, \Delta z)$ for a given dataset are not equal than each coordinate dimension will have a different scaling term. As a result, scalar gradients will be transformed in this new space. If orientation data is available, then the scalar gradients computed from the network are straightforwardly transformed back into the original space. This is required to accurately measure the angular residuals to constrain that type of
data. Since in the present case study orientation data is not used, this aspect was not required. Note that for datasets covering a large geographical area, such as the dataset in this case study, having a constant scaling term for each coordinate is not possible. This is because the $\Delta x, \Delta y$ are multiple orders of magnitude larger than $\Delta z$. Scaling all coordinate dimensions by a





single scaling term for these types of datasets result in negligible variation in $z$ coordinates that are not useful for the network; network's training losses do not decrease with training.


For this case study, the learnable variables of our network, called GeoINR, with model parameters summarized in Table 3 are initialized using the pre-trained model with planar geometry described in Sect. 2.5. These variables are updated using the Adam optimizer within the Pytorch framework so that modelling errors are minimized during training through the backpropagation process. Moreover, the cosine annealing learning rate scheduler was used with this optimizer. The networks

model parameters (Table 3) were established from INR literature (Atzmon and Lipman, 2020; Gropp et al., 2020; Park et al., 2019; Sitzmann et al., 2020; Tancik et al., 2020) and refined through trial and error using various combinations of parameter values. For this case study and other synthetic structural geological models, these parameters produce structurally consistent models with respect to the sampled point constraints. The non-linear activation function used for our network was the parameterized Softplus function


$$\sigma(x) = \frac{1}{\beta} log(1 + e^{\beta x}) \qquad (22)$$

where the parameter $\beta$ controls the variability of modelled interfaces. Smaller values of $\beta$ produce flatter modelled interfaces, whereas higher values produce more locally variant modelled interfaces. Two three-dimensional implicit geological models are produced using these neural network parameters, one using both interface and intraformational points while the other

having just interface points. After training, inference is performed on 4,204,592 points from a voxel grid having 5 km horizontal resolution, and 20 m vertical resolution.





**Table 1**. Interface information.

| Interface | Name | Geological Rule | Series | Unit Above | Unit Below | Above Interfaces | Above Series | Below Interfaces | Below Series |
|---|---|---|---|---|---|---|---|---|---|
| $I_6$ | Lower Paleo Unc | Erosional | $\varphi^4$ | 7 | 6 | n/a | n/a | n/a | n/a |
| $I_5$ | Stonewall | Onlap | $\varphi^3$ | 6 | 5 | $I_6$ | $\varphi^4$ | $I_4, I_3, I_2$ | $\varphi^3, \varphi^3, \varphi^2$ |
| $I_4$ | Stony Mountain | Onlap | $\varphi^3$ | 5 | 4 | $I_6, I_5$ | $\varphi^4, \varphi^3$ | $I_3, I_2$ | $\varphi^3, \varphi^2$ |
| $I_3$ | Red River | Onlap | $\varphi^3$ | 4 | 3 | $I_6, I_5, I_4$ | $\varphi^4, \varphi^3, \varphi^3$ | $I_2$ | $\varphi^2$ |
| $I_2$ | Sub RR Unc | Erosional | $\varphi^2$ | 3 | 2 | $I_6, I_5, I_4, I_3$ | $\varphi^4, \varphi^3, \varphi^3, \varphi^3$ | n/a | n/a |
| $I_1$ | Sub Wpg Unc | Erosional | $\varphi^1$ | 2 | 1 | $I_6, I_5, I_4, I_3, I_2$ | $\varphi^4, \varphi^3, \varphi^3, \varphi^3, \varphi^2$ | n/a | n/a |
| $I_0$ | Precambrian | Erosional | $\varphi^0$ | 1 | 0 | $I_6, I_5, I_4, I_3, I_2, I_1$ | $\varphi^4, \varphi^3, \varphi^3, \varphi^3, \varphi^2, \varphi^1$ | n/a | n/a |

The sequence of above/below interface and series are associated. For example, consider the below interfaces and series for $I_5$. The interface $I_4$ is associated with series $\varphi^3$. Similarly, interface $I_3$ is associated with series $\varphi^3$ and interface $I_2$ is associated with series $\varphi^2$.

**Table 2.** Formation unit information.

| Unit | Name | Series | Unit Above | Unit Below | Above Interfaces | Above Series | Below Interfaces | Below Series |
|---|---|---|---|---|---|---|---|---|
| $U_7$ | Above Youngest (7) | $\varphi^4$ | n/a | 6 | n/a | n/a | $I_6$ | $\varphi^4$ |
| $U_6$ | Interlake (6) | $\varphi^3$ | 7 | 5 | $I_6$ | $\varphi^4$ | $I_5, I_4, I_3, I_2$ | $\varphi^3, \varphi^3, \varphi^3, \varphi^2$ |
| $U_5$ | Stonewall (5) | $\varphi^3$ | 6 | 4 | $I_6, I_5$ | $\varphi^4, \varphi^3$ | $I_4, I_3, I_2$ | $\varphi^3, \varphi^3, \varphi^2$ |
| $U_4$ | Stony Mountain (4) | $\varphi^3$ | 5 | 3 | $I_6, I_5, I_4$ | $\varphi^4, \varphi^3, \varphi^3$ | $I_3, I_2$ | $\varphi^3, \varphi^2$ |
| $U_3$ | Red River (3) | $\varphi^3$ | 4 | 2 | $I_6, I_5, I_4, I_3$ | $\varphi^4, \varphi^3, \varphi^3, \varphi^3$ | $I_2$ | $\varphi^2$ |
| $U_2$ | Winnipeg (2) | $\varphi^1$ | 3 | 1 | $I_6, I_5, I_4, I_3, I_2$ | $\varphi^4, \varphi^3, \varphi^3, \varphi^3, \varphi^2$ | $I_1$ | $\varphi^1$ |
| $U_1$ | Deadwood (1) | $\varphi^0$ | 2 | 0 | $I_6, I_5, I_4, I_3, I_2, I_1$ | $\varphi^4, \varphi^3, \varphi^3, \varphi^3, \varphi^2, \varphi^1$ | $I_0$ | $\varphi^0$ |
| $U_0$ | Precambrian (0) | $\varphi^0$ | 1 | n/a | $I_6, I_5, I_4, I_3, I_2, I_1, I_0$ | $\varphi^4, \varphi^3, \varphi^3, \varphi^3, \varphi^2, \varphi^1, \varphi^0$ | n/a | n/a |






**Table 3.** GeoINR model parameters values.

| Parameters | Value |
|---|---|
| Number of hidden layers $N_h$ | 3 |
| Dimension of representations $d_{rep}$ | 256 |
| Learning rate | 0.0001 |
| Non-linear activation function | Softplus ($\beta = 100$) |
| Number of training epochs | 5000 |
| Global constraint weight $\lambda$ | 0.1 |

**Table 4.** GeoINR model performance metrics.

| Model | Metric | Value |
|---|---|---|
| With intraformational constraints<br>- 4708 interface pts<br>- 11270 intraformational pts<br>- 5000 pts for global constraint | Interface loss $\mathcal{L}_I$ | 0.0119 |
| | Unit loss $\mathcal{L}_U$ | 0.0010 |
| | Global Smoothness loss $\mathcal{L}_\xi$ | 0.0059 |
| | Per epoch training time | 0.4 s |
| | Inference time for voxel grid | 1.4 s |
| | $\overline{\Delta d}_{Train}$ | 6.4 m |
| | $k$-fold (20) $\overline{\Delta d}_{Test}$ | 31.0 m [7.8, 412.1] |
| | $k$-fold (10) $\overline{\Delta d}_{Test}$ | 27.0 m [10.0, 170.2] |
| | $k$-fold (5) $\overline{\Delta d}_{Test}$ | 18.5 m [10.4, 47.8] |
| | $k$-fold (2) $\overline{\Delta d}_{Test}$ | 13.0 m [12.8, 13.1] |
| Without intraformational constraints<br>- 4708 interface pts<br>- 5000 pts for global constraint | Interface loss $\mathcal{L}_I$ | 0.0122 |
| | Global Smoothness loss $\mathcal{L}_\xi$ | 0.0046 |
| | Per epoch training time | 0.2 s |
| | Inference time for voxel grid | 1.4 s |
| | $\overline{\Delta d}_{Train}$ | 6.7 m |
| | $k$-fold (20) $\overline{\Delta d}_{Test}$ | 21.7 m [10.1, 191.4] |
| | $k$-fold (10) $\overline{\Delta d}_{Test}$ | 17.5 m [10.8, 58.1] |
| | $k$-fold (5) $\overline{\Delta d}_{Test}$ | 16.2 m [12.1, 27.7] |
| | $k$-fold (2) $\overline{\Delta d}_{Test}$ | 15.6 m [15.5, 15.7] |

$\overline{\Delta d}_X$ is the mean distance residual between modelled interfaces and pointset $X$, either a training set or a test set. Brackets [] indicate the range of $\overline{\Delta d}_{Test}$ values
from a set generated by each $k$-fold cross-validation procedure.

Results were obtained using a high-end desktop PC with an Intel Core i9-9980XE CPU and a single NVIDIA RTX 2080 Ti
GPU.


The presented data and resulting models in Fig. 8 all have a vertical exaggeration of 100x so that the data and variation of
geological structures can be visualized. For this dataset, the augmented intraformational points provided incremental
refinements to modelled structures. An example of structural refinements is shown on a cross section taken from the middle





portion of the model (Fig. 8c). The model generated without the intraformational units (transparent curves) has the sub-
Winnipeg unconformity (orange) positioned lower than it should since in that location there are no interface points sampling
that unconformity. In addition, there are slight geometry changes for other interfaces with the model using intraformational
units (solid curves) that are attributed to the presence of different units located off section. How the unconformities cut older
stratigraphy and other unconformities can be clearly seen in modelled interfaces and resultant scalar fields in Fig. 8d, along
with the visually impressive data fitting characteristics (also seen in Fig. 8b (left), 8c). The effect of adding the global
smoothness constraint, or Eikonal constraint (Eq. 18), can be seen in a scalar field associated with the youngest unconformity
($\varphi^4$) shown in Fig. 8e (top). Without a global constraint, production of implicit modelling artifacts (e.g., isolated bubbles Fig.
8e (bottom)) can occur when paired with large training epochs and noisy datasets.

Resulting model performance metrics on both datasets used in the case study are summarized in Table 4. These metrics
include loss function values at the last training iteration, computing times, mean distance residuals between modelled interfaces
and associated pointsets, and $k$-fold cross validation results. For computing times, per epoch training times on one GPU led to
a total of ~35 minutes for the model with intraformational constraints and a total of ~20 minutes without when using 5000
training epochs. A larger number of training epochs was chosen to achieve the smallest total error possible. However, even
models generated with 1000 epochs were geologically representative of the basin, but had larger fitting residuals. Note these
computing times can be reduced by simply adding more GPUs and performing distributed training. The tabulated mean
distance residuals, a real-world distance, were computed for the generated models using PyVista (Sullivan and Kaszynski,
2019) to give an intuitive notion of how well the GeoINR network fits the provincial scale dataset. The mean distance residuals
using the whole dataset, $\overline{\Delta d}_{Train}$, was 6.4 meters and 6.7 meters for the model including intraformational constraints and
without, respectively. Most constraints had distance residuals near 0 meters, however some constraints had larger residuals;
some data constraints exhibit a vertical shift upward in comparison to the other wells in the immediate vicinity surrounding
the well. This could be due to faulting, highly variable localized structures, or miss-interpretation. Finally, a systematic $k$-fold
cross validation (Rodríguez et al., 2009) analysis was completed to estimate the prediction error of the GeoINR network where
there is no data available in the modelling domain and assess if the network suffers from overfitting. This analysis involved
splitting the dataset into $k$ partitions or 'folds' and configuring them into $k$ splits. For each split, the GeoINR network is trained
using $k - 1$ of the folds as training data using the same parameters in Table 3. Once the split is trained, the resulting model
is validated on the remaining part of the data, called test data, where the mean distance residuals are computed. The mean
distance residuals on the test data $\overline{\Delta d}_{Test}$ are averaged over all splits and tabulated in Table 4. This procedure was performed
for $k = 20, 10, 5, 2$ and for both with and without intraformational constraints. From these results, it is evident that the GeoINR
network has a reasonably low prediction error, especially given the provincial scale of the geological model and the network
does not suffer from overfitting. See Appendix B for a more detailed summary of the $k$-fold cross validation results.



## 4. Discussion

Our modelling results show that INR networks can be applied to generate basin-scale three-dimensional geological models containing numerous unconformities and conformable stratigraphic interfaces from noisy regional compilation datasets. While

the intraformational constraints only provided incremental improvements to the implicit model for the presented case study, they did serve a purpose in demonstrating they work with the methodology. Furthermore, these types of constraints could provide larger improvements to modelled structures in datasets with much fewer interface constraints, as is commonly found in outcrop datasets where interface observations are rare. In addition, they could provide a mechanism for integrating geological maps in the modelling process, by sampling points within unit polygons and feeding those point constraints in the

modelling and adding a weighting term to those loss functions so that the algorithm does not exactly fit those points but use them as a guide to constrain the model.

The ability of INR networks to process above and below inequality constraints on stratigraphic relations (Eqns. 4, 7, 8), and demonstrated in the case study, show that these networks can be used efficiently to incorporate new knowledge constraints

derived from stratigraphic columns and geological rules (e.g., erosion, onlapping) between various structures. In comparison, classical implicit interpolation methods require solving computationally expensive convex optimizations in order to incorporate inequality constraints with poor scaling as the number of constraints increase. Moreover, these classical methods only apply inequality constraints to a single scalar field, to the best of the authors' knowledge, and not across a series of scalar fields to couple them. Since neural networks do not require solving such expensive optimizations and that they can efficiently

couple a series of scalar fields, they are effective in incorporating the comprehensive suite of knowledge constraints on stratigraphic relations.

Iso-values associated with modelled interfaces in the proposed methodology vary in every training iteration of the learning algorithm so that modelling results are independent of user-defined iso-values. While defining specific interface iso-values is

a straightforward way to encode the stratigraphic sequence (e.g., larger values are younger than smaller values), it is not optimal. Assigning specific iso-values for interfaces heuristically (e.g., uniformly distributed between some numerical range) can negatively impact resulting modelled geometries. This is particularly evident when dealing with different unit thicknesses, varying unit thickness across the modelling domain, and as the number of interfaces increases. With the proposed methodology, the algorithm learns the optimal set of iso-values for the interfaces permitting more complex geological structures to be

modelled. It is important to note that the stratigraphic constraints (Sect. 2.3.1) embed the knowledge of the stratigraphic sequence so that the resulting interface iso-values respect that sequence.

Loss functions used to constrain resulting implicit scalar functions make frequent use of scalar field gradients $\nabla\varphi^i$ computed on pointsets. To compute the gradient of a scalar field generated by an implicit function parameterized by a neural network for



an inputted point, the chain rule is applied to the networks output, $\varphi^i(\boldsymbol{x})$, with respect to the coordinates of the point $\boldsymbol{x} = (x, y, z)$. For machine learning programming frameworks (Pytorch, Tensorflow) this is straightforwardly and efficiently computed in one line of code. Note that higher-order $n$-th derivatives may also be similarly computed (e.g., useful for Laplacian or curvature computations) provided that the non-linear activation function $\sigma$ are at least $n$ times differentiable.

Comparisons of the proposed methodology to the recent GNN deep learning approach (Hillier et al., 2021) for three-dimensional structural geologic modelling will be described. First, the generation of latent representations (e.g., embeddings, features) for MLP networks are at a minimum two orders of magnitudes faster than Hillier et al. (2021). Second, the proposed approach does not require the generation of an unstructured volumetric grid. This requirement prohibits the development of high-resolution models covering large geographical areas. For example, for the modelling domain in the provided case study,

a tetrahedral mesh with varying resolution yielded a tetrahedral mesh requiring ~10 GB of storage, whereas for a voxel grid with high-resolution in the vertical dimension, required only ~150 MB of storage for the points. Lastly, we determined the graph structure, an unstructured mesh, and the associated graph-based convolution operations used in the GNN approach yielded worse modelled geological structures as compared with ones generated using MLP networks; the scalar fields generated by GNN's were much noisier. The reason for this is that the graph edges, edges belonging to faces of tetrahedrons,

did not provide any intrinsic value for improving the networks predictions. For GNN's to provide meaningful improvements to structural modelling, the graph structure must represent meaningful geological concepts, not simply something on which a scalar value is predicted. For example, a graph representing how geological unit volumes are connected to other unit volumes. To do so, graph nodes need to represent regions of the same geological unit or interface, and the edges represent how they are connected to each other as in Thiele et al. (2016). In future research, we would like to derive these types of graphs on the

resulting volumetric geologic unit model produced from the proposed methodology.

In this contribution, we do not tackle important geological discontinuous features such as faults and shear zones commonly found in more complex orogenic and shield terrains. However, since neural networks with similar neural architectures have shown the capacity to approximate discontinuous functions (Lianas et al., 2008, Santa and Pieraccini, 2023), we believe INR

network architectures can support the modelling of these complex features with appropriate enhancements and modifications (e.g., discontinuous activation functions). This is a potential direction for more versatile INR frameworks expanding applications to a wider range of complex geological settings.

Using different activation functions can dramatically affect modelled geometries of geological structures. In this paper, we

used a parameterized Softplus activation function that generated far smoother geometries and improved data fitting as compared to the commonly used ReLU for these networks. ReLU activation functions typically result in modelled geometries with sharp creases in them, which could be potentially more useful in brittle structural settings. Empirically, we tested all





currently available activation functions within the Pytorch framework. The two most reliable activation functions tested experimentally were ReLU and Softplus.


Recent literature on implicit neural representation networks demonstrates that an additional network layer for positional encodings (Tancik et al., 2020) can characterize high frequency components more accurately in modelled outputs. INR networks have been reported as underrepresenting high frequency components of signals and shapes by underfitting these components (Mildenhall et al., 2021). Positional encodings are a common strategy for addressing this issue by transforming

the coordinates of a point into a set of Fourier features which are then feed into the hidden layers of the network. We tested this strategy, and preliminary results indicate that while this technique improved local fitting of high frequency detail when using the ReLU activation function, globally it can generate unsupported large wavelengths of folded features. However, when using the parameterized Softplus function, modelled structures were able to fit the high frequency details sampled by the data constraints while also able to produce robust and geologically representative structures globally without artifacts.

**5. Conclusion**

A methodology is presented that is founded on implicit neural representation networks composed of MLPs for the purposes of three-dimensional implicit geological modelling. The methodology advances existing INR networks used for this purpose by incorporating unconformities into the modelling process, efficient incorporation of new knowledge constraints on stratigraphic relations as well as a global smoothness constraint, and improved training dynamics from the geometrical initialization of

network variables. Combined, these advances permit the modelling of more complex geology and improved data fitting characteristics. In addition, the global smoothness constraint – an Eikonal constraint – provides a mechanism to reduce typical implicit modelling artifacts so that geologically representative models are generated. The presented case study demonstrates that the proposed methodology can generate large regional scale three-dimensional geological models that effectively represent sedimentary basins with unconformities, with notably low fitting residuals and prediction error given noisy compilation

datasets. The possibility to leverage INR networks to support larger-scale implicit geological modelling with big data and geological settings with faults will be studied in future research.







**Appendix A: Iso-surface extraction algorithm**

| Algorithm 1: Cutting iso-surfaces by unconformities |
| --- |

Input:

Scalar field series $\boldsymbol{\varphi}: [N_g, F]$ computed on volumetric grid where $F$ is the number of scalar fields in the series. $N_g$ is the number of grid cell points.

Table $T$ storing associated information for each modelled interface including interface index, scalar field index, iso-value, if is unconformity. Following table constructed using example in Figure 1.

<table>
<tr><td rowspan="2">Interface $I_k$</td><td>$I_k$ interface index $k$</td><td>$\varphi^i$ scalar field index</td><td>$\overline{\varphi}_{I_k^i}^i$ iso-value</td><td>is unconformity</td></tr>
<tr><td>$k$</td><td>$\varphi\_index[k]$</td><td>$Iso[k]$</td><td>$Unc[k]$</td></tr>
<tr><td>5</td><td>3</td><td>$\overline{\varphi}_{I_5^3}^3$</td><td>True</td></tr>
<tr><td>4</td><td>2</td><td>$\overline{\varphi}_{I_4^2}^2$</td><td>False</td></tr>
<tr><td>3</td><td>2</td><td>$\overline{\varphi}_{I_3^2}^2$</td><td>False</td></tr>
<tr><td>2</td><td>1</td><td>$\overline{\varphi}_{I_2^1}^1$</td><td>True</td></tr>
<tr><td>1</td><td>0</td><td>$\overline{\varphi}_{I_1^0}^0$</td><td>False</td></tr>
<tr><td>0</td><td>0</td><td>$\overline{\varphi}_{I_0^0}^0$</td><td>False</td></tr>
</table>

Output: Set of iso-surfaces $S$ cut by unconformities

$N_I = |I|$ (Number of interfaces)

$S = []$ # empty list

$for\ k = 0, \cdots, N_I - 1:$

    # Organize associate scalar field array $\boldsymbol{\varphi}[:, \varphi\_index[k]]$ into 3D grid array $V: [n_x, n_y, n_z]$ , $N_g = n_x \times n_y \times n_z$

    $isosurface\_k = marching\_cubes(V, Iso[k])$

    $S.append(isosurface\_k)$

$for\ k = 0, \cdots, N_I - 1:$

    # Iterate over list of above (younger) unconformity iso-surfaces $y\_unc\_k$ than current iso-surface $S_k$, arranged older to younger

    $for\ y\_unc\ in\ y\_unc\_k:$

        $S_k = cut(S_k, y\_unc)$ # cut $S_k$ surface by $y\_unc$ iso-surface





## Appendix B: *k*-fold cross validation results

**Table B1.** *k*-fold metrics for structural model generated from interface and intraformational constraints.

| *k* | data removed from training (%) | $\overline{\Delta d}_{Test}$ (meters) | Range (meters) |
|---|---|---|---|
| 20 | 5 | 31.0 | [7.8, 412.1] |
| 10 | 10 | 27.0 | [10.0, 170.2] |
| 5 | 20 | 18.5 | [10.4, 47.8] |
| 2 | 50 | 13.0 | [12.8, 13.1] |


**Table B2.** *k*-fold metrics for structural model generated from interface constraints only.

| *k* | data removed from training (%) | $\overline{\Delta d}_{Test}$ (meters) | Range (meters) |
|---|---|---|---|
| 20 | 5 | 21.7 | [10.1, 191.4] |
| 10 | 10 | 17.5 | [10.8, 58.1] |
| 5 | 20 | 16.2 | [12.1, 27.7] |
| 2 | 50 | 15.7 | [15.5, 15.7] |

$\overline{\Delta d}_{Test}$: mean distance residual computed on the testing set (not used to train/fit the model)

See https://scikit-learn.org/stable/modules/cross_validation.html for implementation details and illustration regarding the *k*-fold cross validation procedure. The range of mean distance residuals on test points, $\overline{\Delta d}_{Test}$, from all *k* splits indicate the lower and upper bound of these residuals across all splits for a given *k*. The large mean residual for upper bounds (e.g., 191.4 m for *k* = 20 in Table 6) is the result of a single point constraint associated with the sub-Winnipeg unconformity in the far North-West corner of the modelling domain. For every *k*-fold procedure carried out, there is always one split this point is excluded

from training and results in a larger mean distance residual, and increases with larger *k,* on the corresponding test set. It is important to note that the next nearest constraint to this point associated with this interface (sub-Winnipeg unconformity) is 150 km away. The upper bound on mean distance residuals decreases with smaller *k*. This is because with smaller *k* a higher percentage of constraint points are removed from training and generated models become more generalized. The lower bound of mean distance residuals decreases with larger *k*, since more points are used to constrain generated models.




*Code and data availability*. The source code for the GeoINR neural network developed in Pytorch and data can be freely downloaded from https://github.com/MichaelHillier/GeoINR.git (last access: 28 November 2022) or https://doi.org/10.5281/zenodo.7377977 (Hillier et al., 2022).

*Author contributions*. MH conducted the research, implemented the modelling algorithms, and prepared the manuscript with contributions from all co-authors. FW supervised the research. FW, EdK, BB, ES contributed to the conceptualization of the overarching research objectives and analysis of modelling results from a geological point of view. KB prepared the datasets used for modelling.

*Competing interests*. The authors declare that they have no conflict of interest.

*Acknowledgements*. This research is part of and funded by the Canada 3D initiative at the Geological Survey of Canada. We gratefully acknowledge and appreciate the collaborative work with Arden Marsh and Tyler Music at the Saskatchewan Geological Survey and the GSC for building a three-dimensional geological model of the Western Canadian Sedimentary
Basin in Saskatchewan, Canada, and sharing the dataset used for this research and their extensive geological knowledge in Saskatchewan. Additionally, we are thankful for all the discussions with colleagues from RWTH Aachen University and the Loop 3D project. NRCan contribution number 20220381.

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
