# Peer review of "GeoINR 1.0: an implicit neural network approach to threedimensional geological modelling"

_Geoscientific Model Development, 2022_

## Referee Comment (RC2)

I am delighted to have the chance to review this manuscript. I would like to offer a concise overview of the paper, emphasizing its significant contributions, and also pose a few relevant questions for further consideration. The manuscript presents a novel methodology based on implicit neural representation networks for three-dimensional geological modeling. The methodology advances accurate and detailed geological modeling, particularly for complex subsurface structures. The key contributions include incorporating unconformities, handling complex geological features, integrating stratigraphic constraints, reducing modeling artifacts, and validating the approach through a case study. However, to further enhance the understanding and applicability of the proposed algorithm, addressing the following questions would provide valuable insights.

**1. Modeling Complex Faults:**

The paper introduces an implicit modeling approach that can handle multiple complex faults in a geological model. However, it's essential to understand how the method constructs geological models containing numerous complex faults. How does the methodology define geological domains when faults do not extend throughout the entire model space? Could the authors provide a synthetic or field example to demonstrate this?

**2. Modeling Unconformities:**

The methodology also addresses modeling unconformities, a critical aspect of geology. It's commonly challenging to obtain complete geometric information about subsurface unconformities in practice. How does the proposed method handle scenarios where only sparse and unevenly distributed information about unconformities is available? How are multiple geological domains defined in these cases, and how reliable are the resulting models near unconformity interfaces?

**3. Effect of Loss Functions:**

The paper employs various loss functions to incorporate stratigraphic and structural information during the training of the INR network. Could the authors present a clear example using one of their test datasets to demonstrate how well the trained network fits each individual constraint in the loss functions? Which constraint or loss function has a significant impact on the quality of the modeling results? How are the weights of these multiple loss functions determined, and do they need adjustment for different modeling tasks?

**4. Consistency and Validity of Modeling Results:**

The approach trains a neural network using defined loss functions and then employs the trained network for structural modeling. Can this method guarantee that the modeling results align with known structural features or achieve an exact fit? Additionally, since the loss functions are defined only on scattered points, can the effectiveness of the modeling results be ensured in other regions of the model?

**5. Handling Limited Structural Information:**

A notable strength of the approach is its ability to handle scenarios with limited structural information, often encountered in datasets with rare interface observations, such as outcrop

datasets. Can the method still produce reliable and accurate structural modeling results in such cases? Could the authors provide one or two test examples to illustrate this?

**6. Efficiency and Generalization:**

Do different geological regions require separate network training? Should each geological domain divided by unconformities require individual structural modeling? Could the authors compare the modeling efficiency and accuracy of their learning-based method to traditional implicit modeling approaches, both using the same set of structural data?

**7. Network Architecture Impact:**

It would be helpful if the authors could provide more details about the specific architecture of the neural network used for implicit structural modeling. How does altering the number of network parameters impact the modeling results? How can one select an appropriate neural network architecture for structural modeling tasks when dealing with varying data and complexities?

---

## Author Comment (AC1)

**Authors Response to Refereed Comments for gmd-2022-290**

Dear GMD Referees,

Thank you both for taking the time to thoroughly review our paper entitled "GeoINR 1.0: an implicit neural representation network for three-dimensional geological modelling". We very much appreciate your comments and suggestions for improving the manuscript. We have carefully considered these comments and suggestions and implemented changes according to all the suggestions and submitted my revision of the manuscript.

With kind regards,

Michael Hillier

Note the reviewer's comments (Calibri font) and my comments and argumentation (red Calibri font) how we addressed the issues raised by the reviewers. *Italicized red Calibri font* in quotations is text from the revised manuscript. Referenced page numbers and lines are also from the revised manuscript.

**Reviewer # 1**

"This is a good work, here are some suggestions to the authors."

Thank you very much for appreciating our research. You will find in our revision how we implemented and addressed your suggestions.

**Comment #1**

"In this paper, stacked MLPs are used to build GeoINR. MLPs contain many model parameters. Are the Hidden layer parameters (Nh, activation function, etc.) the same among these stacked MLPs, and how does the initialization of these parameters relate to the complexity of the actual study area?"

Yes, the hidden layer parameters $(N_h, d_{rep}, \sigma)$ where number of hidden layers $N_h$, dimensionality of the hidden representations $d_{rep}$, and activation function $\sigma$ are all the same among the stacked MLPs (P18 361-364). These parameters were first initialized using existing INR research and then refined through trial and error using various combinations of parameter values (P21 L425-433).

We expanded the discussion (P33 L649-662) to provide more detail involving the used architecture including how their parameterization is related to geological complexity:

*"Although the MLP network architecture parameters $(N_h, d_{rep}, \sigma)$ used in this contribution (Table 1) generated reliable and accurate three-dimensional modelling results, the architecture may not be optimal for all geological scenarios. As a general principle, increasing the number of hidden layers $N_h$ tend to improve the capacity of the network to model more complex structures, whereas increasing the dimensionality of representations $d_{rep}$ (number of the neurons in a layer) tend to improve the smoothness of modelled geometries (Hillier et al., 2021). But these effects have diminishing returns as these parameters are further increased. It is important to note that the use of different non-linear activation function σ can dramatically affect modelled geometries. Empirically, we tested all currently available activation functions within the Pytorch framework, and found the two most reliable activation functions were ReLU and Softplus. In this paper, we used a parameterized Softplus activation function that generated far smoother*

*geometries and improved data fitting compared to the commonly used ReLU. ReLU activation functions typically result in modelled geometries with sharp creases, which could be more useful in brittle geological settings. In scenarios where these architectural parameters are not ideal, automated tools are available for optimizing them (Liaw et al., 2018). In general, the best architecture to use for a particular geological scenario is an open research question. This motivates the development of standardized three-dimensional geological models to be used for benchmarking different methods and their parameterizations."*

**Comment #2**

"Can the change curve of the loss function in the training process of the model be given to reflect the training process and effect of the model?"

A similar request was made by reviewer #2. Loss function curves for every model generated in the manuscript are added to show training dynamics and the relative impacts of individual loss functions and associated constraints. For case study 1, this is presented in Figure 10. Furthermore, observations on these plots are provided in the text (P27 L525-530). For case study 2, this is presented in Figure D1 (Appendix D since there were many plots for 5 different models) and plot observations are given in the text (P40 L773-776).

**Comment #3**

"Line 98. The format of the paragraph is not consistent with other paragraphs."

Fixed – indented this paragraph as required.

**Comment #4**

"The resolution of some pictures in the paper is low, resulting in unclear details in the pictures (Figure 1 and Figure 8)."

Higher resolution pictures are provided with the revised manuscript. Although the pdf version of the manuscript may reduce to resolution of pictures in our Microsoft Word document, GMD has the high-resolution pictures that will be used for publication.

**Comment #5**

"Punctuation marks after formulas are not in uniform format."

Fixed – added punctuation were needed throughout Section 2 where there are formulas.

**Comment #6**

"The layout of Figure 9 is not consistent with other figures."

Fixed – changed sampled intraformational points represented by triangles with smaller circles (e.g., smaller than the circles representing interface points). It is now consistent with other figures.

**Reviewer # 2**

"I am delighted to have the chance to review this manuscript. I would like to offer a concise overview of the paper, emphasizing its significant contributions, and also pose a few relevant questions for further consideration."

Thank you very much for taking the time to provide a thorough review of our manuscript. We appreciate this detail greatly as it did indeed provide valuable insight that allowed us to improve the manuscript.

**Comment #1 (Modeling Complex Faults)**

"The paper introduces an implicit modeling approach that can handle multiple complex faults in a geological model. However, it's essential to understand how the method constructs geological models containing numerous complex faults. How does the methodology define geological domains when faults do not extend throughout the entire model space? Could the authors provide a synthetic or field example to demonstrate this?"

As indicated in the manuscript (P20 L392-L394, P33 L675-679, P34 L692-693) the handling of faults is not considered in this work, but to be incorporated in future research. The handling of faults would be itself a separate paper dedicated to this topic. Please note for the sedimentary basin case study, faults are not significant for the provincial scale of the geological model produced. In future research, our aim is not to use the idea of geological domains for the hanging/foot wall but instead learn discontinuous functions modelling the implicit scalar field. This way, the method could support dying/terminating faults and structures sampled on one side of the fault that could influence/constrain the geometry on the other side of the fault. In addition, this strategy would be more computationally efficient - reduction in the number of scalar fields required to be modelled/learned by the network.

**Comment #2 (Modeling Unconformities)**

"The methodology also addresses modeling unconformities, a critical aspect of geology. It's commonly challenging to obtain complete geometric information about subsurface unconformities in practice. How does the proposed method handle scenarios where only sparse and unevenly distributed information about unconformities is available? How are multiple geological domains defined in these cases, and how reliable are the resulting models near unconformity interfaces?"

The proposed method can well handle sparse and unevenly distributed data in geological settings containing unconformities. This has been demonstrated for both case studies (included additional case study in revised manuscript) that have sparse and scattered points sampling unconformities. For the basin dataset (case study 1), the unevenly distributed, and sparse unconformity observations in some localized areas are shown on the left-hand side of Fig. 8a in the north and northeastern parts of the model. For the outcrop dataset (case study 2), Figure 12 illustrates the data's degree of sparsity and uneven spatial distribution. There are only 3 points sampling the unconformity at the topographic surface in this case study.

In the manuscript, how geological domains are defined is the same for any type of data configuration (e.g., dense, sparse, regular, irregular). This is described in detail in Section 2.6 and illustrated in Figure 7. To quickly summarize the process:

They are defined by applying Boolean masks associated with unconformity interfaces that are sampled. Each geological domain has a unique set of operations using these Boolean masks defined from the geological history/stratigraphic column. For example, the youngest geological domain is above the youngest unconformity interface (mask is True wherever in the model space that the scalar field is >= the iso-value associated with the youngest unconformity), whereas the oldest geological domain is defined by a set of operations using all the boolean masks - each boolean mask can potentially remove (erode) volumes of the model space if a younger unconformities surface intersects the geometry of older modeled interfaces. Please note that unconformities can erode anything that is older and importantly, the geometry of all the modelled interfaces including unconformities (and associated geological domains) is updated/changes every iteration of the learning algorithm until training stops/convergence is achieved.

"How reliable (e.g., uncertainty) are the results near unconformities?"

If this question is regarding the how predictive the generated implicit model is:

The reliability of the results depends on multiple factors: 1) the presence of nearby data constraints (unit, interface, and unconformity markers), 2) geometric variability of subsurface structures (complexity), 3) thickness of units above/below near unconformity, and 4) supplied knowledge constraints (stratigraphic column).

If the question is regarding the jagged (e.g., saw tooth) behaviour of modelled unconformities surfaces in many other implicit modelling codes our modelled interface surfaces do not suffer from this (see Sect. 2.7). This is because we perform the iso-surface extraction on the continuous scalar fields, then cut these continuous surfaces using the appropriate geological rules according to the stratigraphic column to generate discontinuous surfaces without these jagged features.

Finally, three-dimensional geological models generated by the proposed method for both case studies containing unconformities (sparsely and unevenly sampled) were compared to other models (hydrid implicit/explicit approach, and geological map) to clearly demonstrate the reliability of the GeoINR produced models – illustrated in new figures (Fig. 11, 12).

**Comment #3 (Effect of Loss Functions)**

"The paper employs various loss functions to incorporate stratigraphic and structural information during the training of the INR network. Could the authors present a clear example using one of their test datasets to demonstrate how well the trained network fits each individual constraint in the loss functions? Which constraint or loss function has a significant impact on the quality of the modeling results? How are the weights of these multiple loss functions determined, and do they need adjustment for different modeling tasks?"

A similar comment was asked by reviewer #1. Loss function curves for every model generated in the manuscript were added to show training dynamics and the relative impacts of individual loss functions. For case study 1, this is presented in Figure 10. Furthermore, observations on these plots are provided in the text (P27 L525-530). For case study 2, this is presented in Figure D1 (Appendix D) and plot observations are given in the text (P40 L773-776). To quickly summarize: Early training epochs are significantly impacted by the stratigraphic relation constraints if no orientation data is used, otherwise the orientation constraint is by far the driving factor in training dynamics. In later training epochs, the *on* stratigraphic relation

constraint and to a lesser extent the global smoothness constraint (if used) are the primary sources of modelling errors (network losses).

The only loss function that is weighted is the global smoothness term. Empirically on all the datasets used, the other loss functions do not need to be weighted differently to get reasonable results. However, this isn't to say there may be some scenarios where it may be warranted. If this was the case, then an MTL (Multi-Task-Learning) strategy could be employed. Here, the network would dynamically determine what the weights would be (note: the weights would be variables that change every training iteration).

**Comment #4 (Consistency and Validity of Modeling Results)**

"The approach trains a neural network using defined loss functions and then employs the trained network for structural modeling. Can this method guarantee that the modeling results align with known structural features or achieve an exact fit? Additionally, since the loss functions are defined only on scattered points, can the effectiveness of the modeling results be ensured in other regions of the model?"

*Alignment to structural features*

The method would guarantee that the modelling results align with known structural features sampled by data constraints. Furthermore, in our revision we have validated modelling results with other models of the same structural features produced by a hybrid implicit-explicit approach in Gocad and a recent geological map for case studies 1 (Fig. 11) and 2 (Fig. 12), respectively. In both case studies, models generated with the proposed method well align with the known structural features.

*Exact fitting*

In theory (Hornik et al 1989), MLPs can be universal approximators for any function if there are enough hidden neurons (P8 L176-178). For synthetic examples it was possible to obtain an exact fit. However, when using real world data (noisy, imprecise interpretative data) and using multiple loss functions, exact fit (total loss = 0) is improbable. I will emphasize that exacting fitting means the model fits the data within machine precision (~1E-7 floating point). It is possible for models to look as an exact fit but isn't numerically. From case study 2, it is certainly possible to get 'close' (total loss after training < 1E-4) to exact fitting when used data points are geologically consistent with nearby data points (P31 L605-607). However, the more noisy or conflicting data that exists, and to some degree geological complexity, the larger the errors/residuals will be.

*Effectiveness of results in other regions with no data.*

The effectiveness of the modelling results can be ensured in other regions of the model away from data through the global smoothness term. This was thoroughly tested in case study 1 using the k fold cross validation technique (P27 L541-550) where you randomly remove data from training and evaluate the GeoINR model there and compare to the full dataset. It consistently made excellent predictions (low prediction error) in these regions of the model.

**Comment #5 (Handling Limited Structural Information)**

"A notable strength of the approach is its ability to handle scenarios with limited structural information, often encountered in datasets with rare interface observations, such as outcrop datasets. Can the method

still produce reliable and accurate structural modeling results in such cases? Could the authors provide one or two test examples to illustrate this?"

Our developed methodology can support not only basin datasets rich with interface observations, but also outcrop datasets where interface observations are rare and moderate geological unit observations. The method has a unique advantage over any other existing method for this type of dataset – it can efficiently process geological unit observations and rich stratigraphic relationship knowledge. Since this is the case, we decided to add a case study to highlight this ability in section 3.2 in the revised manuscript. Modelling results demonstrate that the proposed method can reliably and accurately generate 3D geological models in outcrop datasets.

As mentioned before, in general, model performance (reliability and data fitting) always depends on multiple factors (geological complexity, data availability, and data quality (robust and consistent interpretations).

**Comment #6 (Efficiency and Generalization)**

"Do different geological regions require separate network training? Should each geological domain divided by unconformities require individual structural modeling? Could the authors compare the modeling efficiency and accuracy of their learning-based method to traditional implicit modeling approaches, both using the same set of structural data?"

In our neural architecture, each MLP represents an implicit scalar function that models a distinct geological feature (conformal stratigraphic package, unconformity interface) that each requires training. It is important to note that the variables of one MLP can influence/constrain other MLPs (and vice versa) because stack MLPs are coupled through the *above/below* stratigraphic relation constraints.

The proposed method can not be compared to traditional implicit modelling approaches because it is not computational possible. First, traditional approaches can not process the large number of inequality constraints (stratigraphic relationships) introduced in the manuscript. Our early work, GRBF (Hillier et al., 2014), we introduced a mathematical framework to incorporate only above/below inequalities for a single continuous interface with two units. Processing more than a thousand of these constraints will take exceeding amounts of compute time (e.g., greater than week(s)) due to costly quadratic optimizations that simply do not scale well. Furthermore, there are no other methods that can incorporate the comprehensive suite of inequalities derived from stratigraphic columns with unconformities, as presented in the manuscript. In addition, the global smoothness constraint can not be incorporated into traditional methods, because of the norm of the gradient of the scalar field which can only be computed after you have a solution. In theory, one could design some sort of iterative method to deal with this complication, but it would require significant research to resolve.

To address the question of the accuracy of our learning-based method, for both case studies we validated our generated models against other types of geological models of the same structural features – albeit not generated by purely traditional implicit methods using the same dataset – since no other method can generate models using the same input. Since this is the case, the next best solution is to compare to other types of geological models that have been validated by geological experts. For case study 1, a recently published hydrid implicit-explicit model was used, whereas for case study 2 a published geological map

was used. In both case studies, models generated by GeoINR aligned well with known geological structures as shown in Figure 11 and 12.

**Comment #7 (Network Architecture)**

"It would be helpful if the authors could provide more details about the specific architecture of the neural network used for implicit structural modeling. How does altering the number of network parameters impact the modeling results? How can one select an appropriate neural network architecture for structural modeling tasks when dealing with varying data and complexities?"

A similar question was also asked by reviewer #1. While these questions are partially addressed in our earlier GNN paper (2021) we expanded the discussion to include more of these details.

P33 L649-662:

*"Although the MLP network architecture parameters $(N_h, d_{rep}, \sigma)$ used in this contribution (Table 1) generated reliable and accurate three-dimensional modelling results, the architecture may not be optimal for all geological scenarios. As a general principle, increasing the number of hidden layers $N_h$ tend to improve the capacity of the network to model more complex structures, whereas increasing the dimensionality of representations $d_{rep}$ (number of the neurons in a layer) tend to improve the smoothness of modelled geometries (Hillier et al., 2021). But these effects have diminishing returns as these parameters are further increased. It is important to note that the use of different non-linear activation function $\sigma$ can dramatically affect modelled geometries. Empirically, we tested all currently available activation functions within the Pytorch framework, and found the two most reliable activation functions were ReLU and Softplus. In this paper, we used a parameterized Softplus activation function that generated far smoother geometries and improved data fitting compared to the commonly used ReLU. ReLU activation functions typically result in modelled geometries with sharp creases, which could be more useful in brittle geological settings. In scenarios where these architectural parameters are not ideal, automated tools are available for optimizing them (Liaw et al., 2018). In general, the best architecture to use for a particular geological scenario is an open research question. This motivates the development of standardized three-dimensional geological models to be used for benchmarking different methods and their parameterizations."*